# Development of circulating isolates of *Plasmodium falciparum* is accelerated in *Anopheles* vectors with reduced reproductive output

Kristine Werling[1], Maurice A. Itoe[1], W. Robert Shaw[1,2], Raymond Dombagniro Hien[3], Bali Jean Bazié[3], Fofana Aminata[3], Kelsey L. Adams[1], Bienvenu Seydou Ouattara[3], Mathias Sanou[3], Duo Peng[1], Roch K. Dabiré[3], Dari F. Da[3], Rakiswendé Serge Yerbanga[3], Abdoulaye Diabaté[3], Thierry Lefèvre[3,4]*, Flaminia Catteruccia[1,2]*

1 Department of Immunology and Infectious Diseases, Harvard T.H. Chan School of Public Health, Boston, Massachusetts, United States of America, 2 Howard Hughes Medical Institute, Chevy Chase, Maryland, United States of America, 3 Institut de Recherche en Sciences de la Santé/Centre Muraz, Bobo-Dioulasso, Burkina Faso, 4 MIVEGEC, IRD, CNRS, University of Montpellier, Montpellier, France

* thierry.lefevre@ird.fr (TL); fcatter@hsph.harvard.edu (FC)

**Data Availability Statement:** The datasets generated and analyzed in the current study are

## Abstract

*Anopheles gambiae* and its sibling species *Anopheles coluzzii* are the most efficient vectors of the malaria parasite *Plasmodium falciparum*. When females of these species feed on an infected human host, oogenesis and parasite development proceed concurrently, but interactions between these processes are not fully understood. Using multiple natural *P. falciparum* isolates from Burkina Faso, we show that in both vectors, impairing steroid hormone signaling to disrupt oogenesis leads to accelerated oocyst growth and in a manner that appears to depend on both parasite and mosquito genotype. Consistently, we find that egg numbers are negatively linked to oocyst size, a metric for the rate of oocyst development. Oocyst growth rates are also strongly accelerated in females that are in a pre-gravid state, i.e. that fail to develop eggs after an initial blood meal. Overall, these findings advance our understanding of mosquito-parasite interactions that influence *P. falciparum* development in malaria-endemic regions.

## Author summary

Malaria, an infectious disease caused by *Plasmodium* parasites, continues to affect millions of people each year. These parasites are transmitted by *Anopheles* mosquitoes during blood feeding, which is required by females to produce eggs. After they are ingested by the female, parasites start developing in the midgut at the same time as the mosquito initiates oogenesis, the process of egg development. Here, we investigated how the concomitant processes of egg and parasite development interact in field-derived *Anopheles* and *Plasmodium* isolates. We found that in these naturally occurring pairs, oogenesis is negatively linked to parasite development, such that mosquitoes producing fewer eggs harbor faster

available in the Harvard Dataverse repository under the identifier https://doi.org/10.7910/DVN/AVVLVI.

**Funding:** This work was funded by the National Institutes of Health (NIH) (R01 AI124165, R01 AI104956) and by a Faculty Research Scholar Award by the Howard Hughes Medical Institute (HHMI) and the Bill & Melinda Gates Foundation (BMGF) (Grant ID: OPP1158190) to FC. FC is a HHMI Investigator. The funders had no role in the study design, data collection, data analysis, data interpretation, decision to publish or preparation of the manuscript. The findings and conclusions herein do not necessarily reflect positions or policies of the NIH, HHMI or BMGF.

**Competing interests:** The authors have declared that no competing interests exist.

growing oocysts. Furthermore, when we disrupt signaling pathways that drive oogenesis, oocyst growth is accelerated in a manner that appears to depend on both mosquito and parasite genotype. Additionally, when field mosquitoes fail to develop eggs, they are likely to have faster growing oocysts. The interaction between key mosquito and parasite developmental processes revealed here underscores how intimately linked *Plasmodium* are to mosquito physiology and can help inform the development of future vector control tools that are both safe and effective.

## Introduction

*Plasmodium* parasites, the causative agents of malaria, are transmitted by female mosquitoes from the *Anopheles* genus. Like all anautogenous mosquitoes, female *Anopheles* require a blood meal to produce eggs, and parasites take advantage of this requirement to enable their transmission between human hosts. Blood feeding triggers a vast metabolic and transcriptional program, largely regulated by the insect steroid hormone 20-hydroxyecdysone (20E), which ultimately results in the production of an egg batch [1–3]. Concurrently, ingested parasites begin their development in the mosquito gut when male and female gametes fuse to form a zygote, which, by about 20 hours post-infection [4], transforms into a motile ookinete that breaks across the midgut epithelium and settles beneath the basal lamina to form a sessile oocyst. Oocysts subsequently grow over several days and through asexual replication (sporogony) produce thousands of sporozoites, which are eventually released into the mosquito hemolymph and invade the salivary glands, from where they can be transmitted to the next human host when the female feeds on blood again.

One critical aspect of *Plasmodium* infection in the mosquito is the rate at which oocysts develop, as this ultimately determines the extrinsic incubation period (EIP) [5], or the length of time it takes parasites to complete sporogonic development and reach transmission stages. The EIP is a key determinant of malaria transmission due to the lengthy parasite developmental cycle. Indeed, *Plasmodium falciparum*, the deadliest form of human malaria, requires a minimum of 9 days for development [6–10], mostly due to the extensive time needed for oocyst growth. This time constitutes a significant fraction of the rather limited lifespan of female mosquitoes in the field, which is variable [11,12] but estimated to last for 2–3 weeks [13–15].

Factors known to affect the length of *P. falciparum* sporogony are temperature [16–19], adult diet [20], larval nutrition [21], and multiple blood meals [8,22–25]. Additionally, we have demonstrated that the rate of *P. falciparum* development in *Anopheles gambiae*—a major malaria vector across much of sub-Saharan Africa—is also linked to the hormonally-regulated processes leading to oogenesis. Impairing the mosquito's ability to produce eggs through several methods, including disrupting the function of the steroid hormone 20E via silencing its heterodimeric nuclear receptor *Ecdysone Receptor* (*EcR*)/*Ultraspiracle*, resulted in faster oocyst development. This in turn resulted in sporozoites reaching the salivary glands sooner while, crucially, still maintaining full infectivity to human hepatocytes [26]. Moreover, we determined that egg numbers are negatively correlated to *P. falciparum* growth rates, a finding implying that mosquitoes with reduced reproductive output can transmit parasites sooner, thereby providing parasites with a significant advantage given the short mosquito life span. Understanding whether this interaction is relevant in malaria-endemic regions where different *P. falciparum* genotypes and multiple *Anopheles* species coexist is essential to reveal the complex dynamics of this vector-parasite interplay.

Here, we examine interactions between oogenesis and sporogony in natural settings in Burkina Faso using multiple *P. falciparum* isolates collected from gametocyte donors and field-derived colonies of two *Anopheles* species. We determine that silencing the 20E nuclear receptor *EcR* in *An. gambiae* and its sibling species *Anopheles coluzzii* leads to faster parasite development. In agreement with these findings, in both species, we find that oocyst growth rates are negatively linked to oogenesis such that females naturally producing fewer eggs harbor faster developing oocysts. We observe variability in the oocyst growth response to *EcR* silencing across different *P. falciparum* isolates and between the two *Anopheles* species, suggesting that both parasite and mosquito genotype are determinants of growth. Additionally, parasite growth rates are significantly accelerated in field-collected *An. coluzzii* females that are in a pre-gravid state (i.e. unable to develop eggs after an initial blood meal), a very common phenomenon in natural mosquito populations [27,28]. Overall, our data across diverse *Anopheles-Plasmodium* combinations suggest that parasite development is favored in females that have reduced or impaired oogenesis, expanding our understanding of factors affecting sporogony in the field. These data also have implications for malaria control strategies that aim to interfere with mosquito reproduction.

## Results

### Impairing 20E signaling leads to accelerated *P. falciparum* oocyst growth in both *An. coluzzii* and *An. gambiae*

To investigate if *P. falciparum* development is affected by *Anopheles* oogenesis in field settings, we performed infection experiments in Burkina Faso with naturally circulating *P. falciparum* isolates collected from six gametocyte carriers, here denoted as patient 1–6 (p1-p6), without culture adaptation [29]. We used colonies of *An. coluzzii* and *An. gambiae* that are frequently replenished from local wild populations. In both species, we impaired 20E signaling via RNAi silencing of the ecdysone nuclear receptor (ds*EcR*), or *green fluorescent protein* (ds*GFP*) as a control. Three *P. falciparum* isolates were used to infect *An. coluzzii* (p1-p3) (**Fig 1A**), and five were used to infect *An. gambiae* (p1, p3-p6) (**Fig 1B**). Although our initial aim was to perform infections in both mosquito species using all parasite isolates, this proved impossible due to the inherent limitations of working with human-collected parasites and field-derived mosquito colonies and due to the fact that gametocyte-containing isolates cannot be stored and used for mosquito infections on a later date. We could therefore conduct only two infections where the same parasite isolates (p1 and p3) were tested side by side in both *An. gambiae* and *An. coluzzii*, and three infections where different parasite isolates were tested in parallel in the same mosquito species (p1-p2 in *An. coluzzii*, and p3-p4 and p5-p6 in *An. gambiae*), allowing direct comparisons in these instances (**Fig 1A and 1B**).

We first assessed egg development in these infected females and determined that while eggs were not significantly affected by ds*EcR* treatment in *An. coluzzii* (**Figs 1C and S1A**), they were significantly decreased in *An. gambiae* following ds*EcR* injections (**Figs 1D and S1B**). We next counted oocyst numbers and found that *EcR* silencing had no effect on the prevalence or intensity of infection in either species (**Figs 1E–1F and S2A–S2B**), partially in contrast to our laboratory observations which showed a reduction in oocyst numbers but also no change in oocyst prevalence for *An. gambiae* in these conditions [26]. We went on to determine parasite growth rates by measuring the cross-sectional area of oocysts at 8 days post-infected blood meal (pIBM) [26] and found a striking increase following ds*EcR* injections in both *Anopheles* species. Specifically, in EcR-depleted females *P. falciparum* oocysts were 55% larger in *An. coluzzii* and 78% larger in *An. gambiae* as compared to their respective ds*GFP* controls (**Fig 2A–2C**).

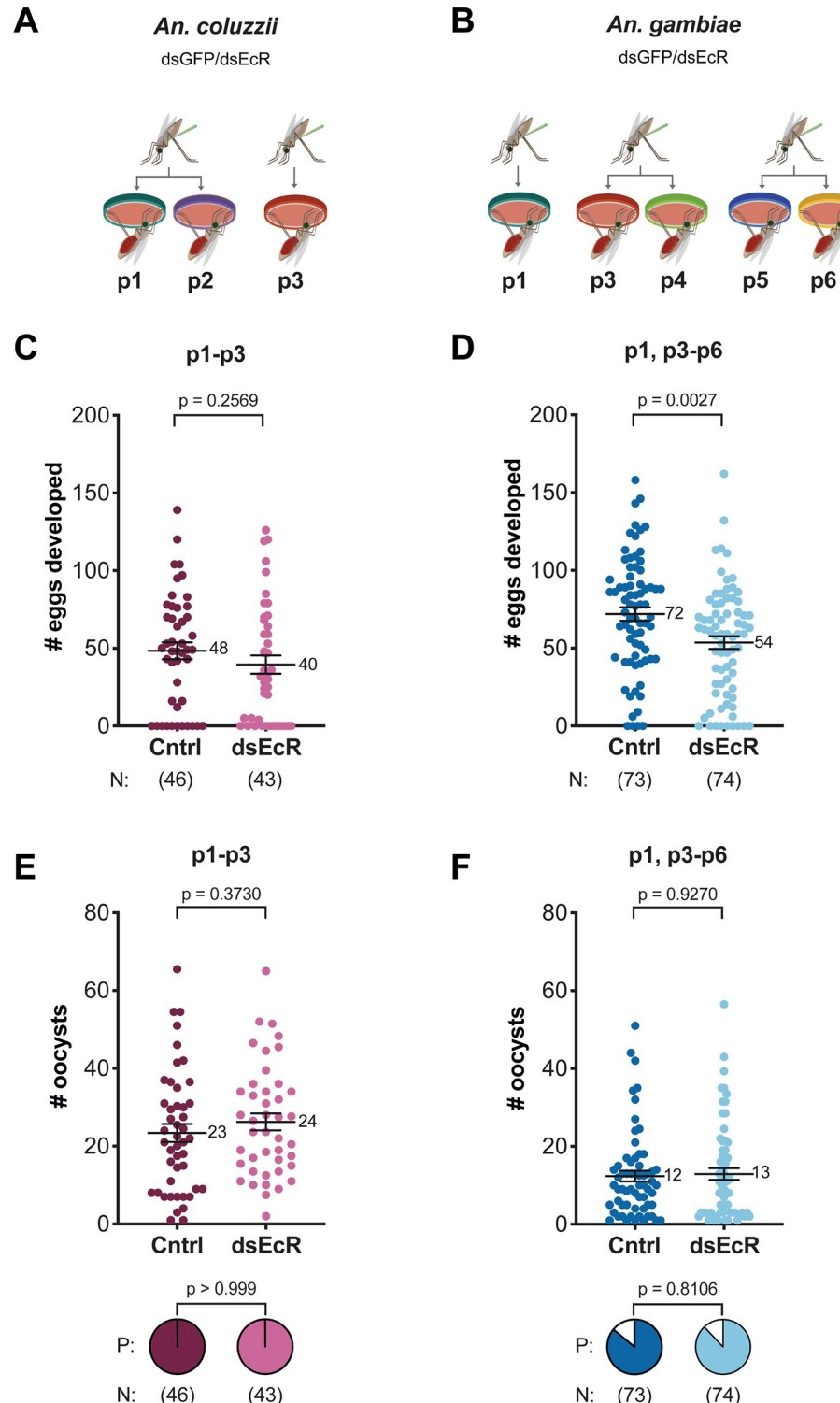

**Fig 1. *An. coluzzii* and *An. gambiae* females were infected with *P. falciparum* field isolates following *EcR*-silencing. (A-B)** *An. coluzzii* and *An. gambiae* females were injected with either ds*GFP* (Cntrl) or ds*EcR* and then infected with *P. falciparum* field isolates originating from six different gametocyte carriers (designated patient, p#). **(A)** Two batches of *An. coluzzii* females were infected with parasites from p1, p2, and p3. **(B)** Three batches of *An. gambiae* females were infected with parasites from p1, p3, p4, p5, and p6. **(C)** In *An. coluzzii*, ds*EcR*-injections did not reduce cumulative egg development to the level of statistical significance (Mann-Whitney). **(D)** In *An. gambiae*, *EcR*-silencing did result in a significant reduction in cumulative egg numbers compared to controls (unpaired t-test). **(E-F)** In both

species, (**E**) *An. coluzzii* and (**F**) *An. gambiae*, ds*EcR*-injections had no effect on cumulative oocyst prevalence (Fisher's Exact) or intensity (unpaired t-test and Mann-Whitney). P next to pie charts = prevalence. N = sample size. p# = parasite isolate.

Our studies using cultured NF54 parasites had shown that larger oocysts are indicative of faster parasite development [8,26]. Although we did not measure EIP directly in this study, we found that ds*EcR An. gambiae* females had a 5.5-fold increase in the number of sporozoites in their salivary glands 12 days after infection relative to controls (**Figs 2D and S3**). Notably, in our experimental settings [26], 12 days pIBM is a time point when sporozoites have started but not yet completed invasion of the salivary glands, so it is likely that this increase in sporozoites reflects faster parasite development, also considering the absence of any difference in oocyst numbers between conditions.

To determine both the successful occurrence of *EcR* silencing (which is difficult to prove given the low expression levels of this nuclear receptor in the midgut) and a possible mechanism behind accelerated growth, we measured expression levels of the lipid transporter *Lipophorin* (*Lp*). In previous studies in the laboratory, we observed that ds*EcR* injections induced upregulation of *Lp*, in turn facilitating faster oocyst growth rates [26]. In ds*EcR* females of *An. coluzzii* and *An. gambiae* we detected elevated expression of this lipid transporter (**Fig 2E**), confirming *EcR* depletion and suggesting a greater availability of lipids to the developing oocysts, as observed in laboratory studies [26]. Combined, these data show that when 20E signaling is disrupted, growth of *P. falciparum* field isolates is accelerated, and this effect may be mediated by Lp-transported lipids, consistent with previous findings [26].

## Parasite growth is negatively linked to egg development via 20E signaling

We next determined the relationship between oocyst size and egg numbers within our two *Anopheles* species. To do this, we used a Generalized Linear Mixed-Effects Model (GLMM) [30,31] to analyze paired egg-oocyst data collected from all *P. falciparum*-infected females (**S1 and S2 Table**). In *An. coluzzii*, there was a significant two-way interaction between egg number and ds*EcR* treatment (**S1 Table**), indicating that the effect of egg development on oocyst size differs between control and ds*EcR* females. Consistent with this, when analyzed independently, while *An. coluzzii* controls showed a significant negative association between egg numbers and oocyst size, in *EcR*-silenced females this association was lost (**Fig 3A**). These findings mean that in *An. coluzzii*, oocysts tend to be larger in females that naturally produce fewer eggs, and conversely, females that develop more eggs are likely to have smaller oocysts, but impairing 20E-signaling breaks this link between egg and oocyst development.

Instead in *An. gambiae* we found a complex three-way interaction between egg number, ds*EcR* treatment, and oocyst number (**S2 Table**), suggesting that both 20E signaling and oocyst density regulate the egg-oocyst relationship. When analyzed independently, both control and *EcR*-silenced females have a general negative association between egg numbers and oocyst size (**Fig 3B**), and this association is strongest at low (<10) oocyst densities in controls and at high (>30) oocyst densities in ds*EcR* females (**Fig 3C**). As with *An. coluzzii*, these findings demonstrate that oocyst growth is inversely linked to a female's investment in reproduction, likely because resources that are not utilized for oogenesis become available to parasites. All together these data reveal that egg development is an important factor affecting oocyst growth in two *Anopheles* species, but also, that signaling by the steroid hormone 20E may differently regulate the egg-oocyst interaction in these mosquito genetic backgrounds.

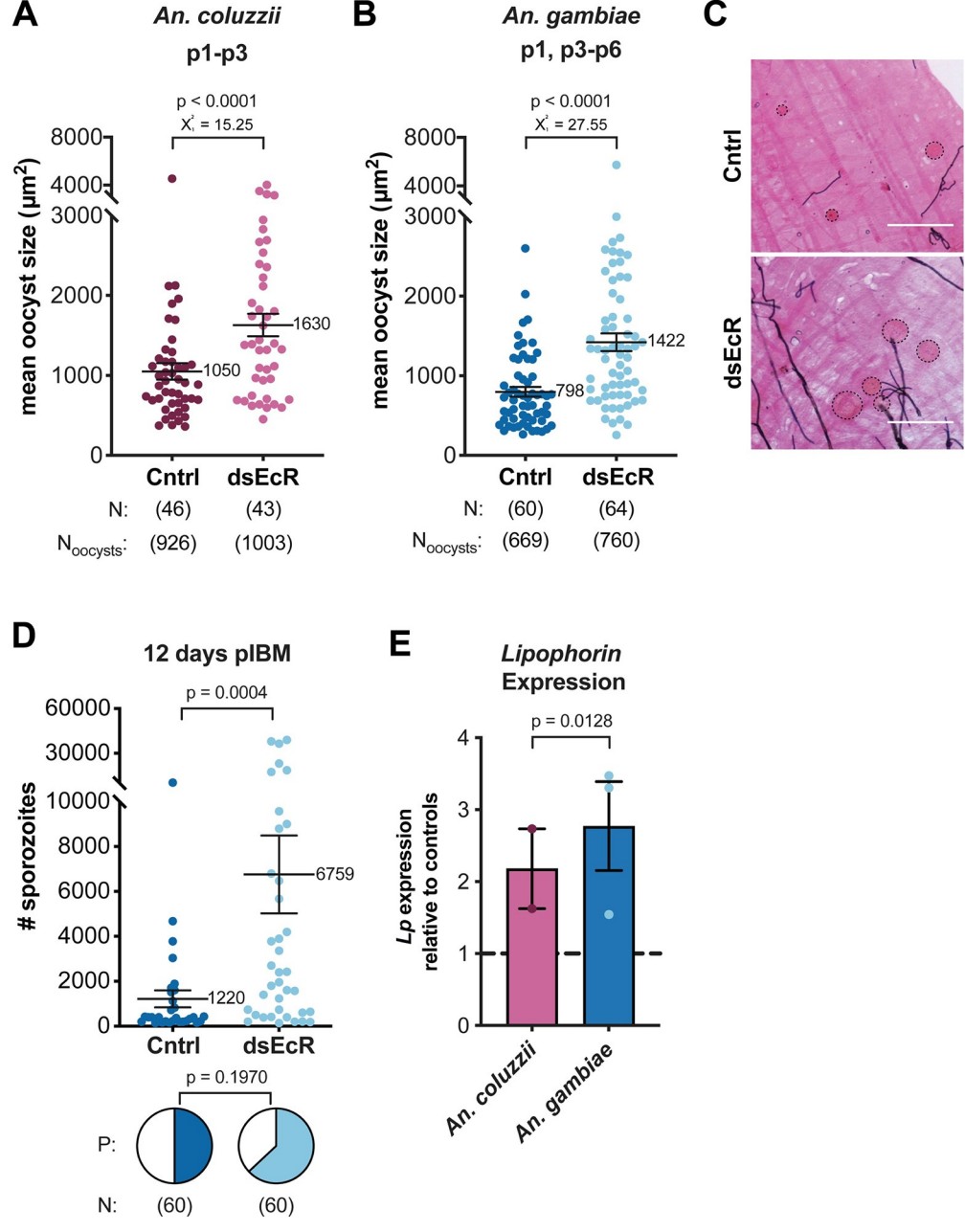

**Fig 2. Oocyst growth is accelerated after ds*EcR* treatment in both *An. coluzzii* and *An. gambiae*. (A)** In *An. coluzzii* and **(B)** *An. gambiae*, *EcR*-depleted females had significantly larger oocysts at 8 days pIBM compared to ds*GFP* controls (Cntrl) (GLMM, LRT). Average oocyst size per midgut is shown for simplicity–analyses are based on nested data incorporating all oocyst measurements. **(C)** Representative images of oocysts found in the midgut of *An. gambiae* control and ds*EcR* females. Black dotted circles show oocyst perimeters. Scale bar = 100 μm. **(D)** At 12 days pIBM, *EcR*-silenced *An. gambiae* females had a greater number of sporozoites in their salivary glands than controls (Mann-Whitney). Sporozoite prevalence (P) at this time point was not different (Fisher's Exact). **(E)** Expression of *Lipophorin* in ds*EcR* females was elevated relative to ds*GFP* controls (unpaired t-test). Gene expression was assessed for each batch of mosquitoes used in infections (whole body mosquito sample without head, prior to blood feeding). For applicable panels, N = sample size, or number of mosquitoes. $N_{oocysts}$ = number of individual oocyst measurements. p# = parasite isolate.

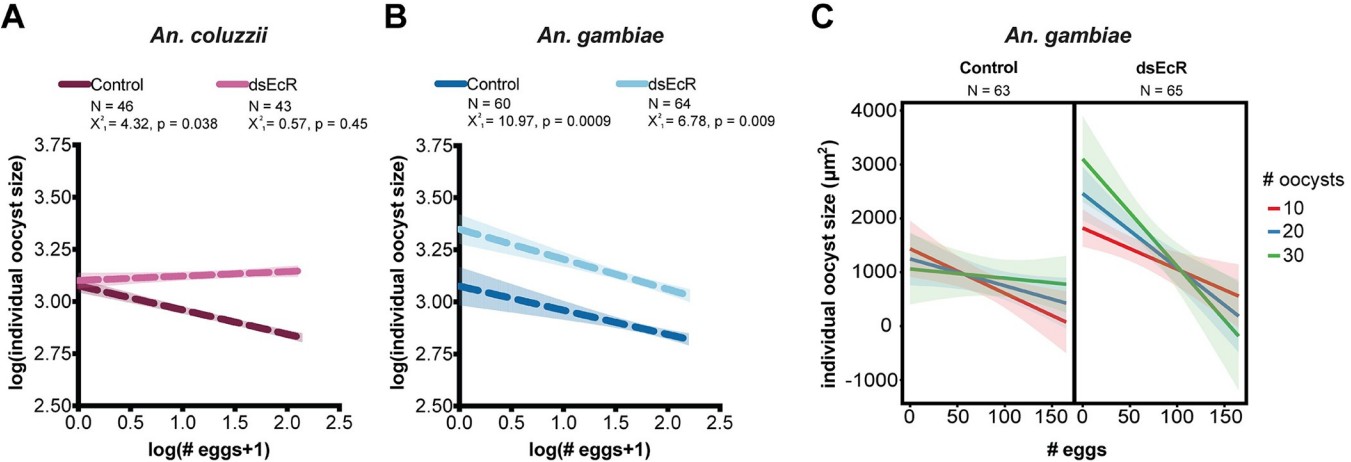

**Fig 3. *P. falciparum* oocyst growth is negatively linked to egg development. (A)** In *An. coluzzii* controls (ds*GFP*-injected), egg numbers are negatively associated with oocyst size, but this association is lost following ds*EcR* treatment (GLMM, LRT). **(B)** In *An. gambiae*, egg numbers are negatively associated with oocyst size in both control and ds*EcR* conditions (GLMM, LRT), but **(C)** this association differentially varies across oocyst density in control and ds*EcR* females (3-way interaction, treatment*egg#*oocyst#, GLMM, LRT, $X^2_1 = 8.57$, p = 0.003). Lines across egg numbers and oocyst size graphically represent the model-based analysis that was performed, which used nested individual oocyst measurements. Shading shows 95% confidence interval. N = sample size, or number of mosquitoes. Number of individual oocyst measurements including in analysis were: *An. coluzzii* controls = 926, *An. coluzzii* ds*EcR* = 1003, *An. gambiae* controls = 669, *An. gambiae* ds*EcR* = 760.

## Parasite isolates vary in their growth response to impaired 20E signaling

As malaria is highly prevalent in the villages surrounding Bobo-Dioulasso [32,33], we reasoned that individual patients may have polyclonal infections comprised of multiple *P. falciparum* genotypes. Indeed, we found that most of the isolates used in our infections contain two or more distinct parasite genotypes, as determined by analyzing the *merozoite surface protein 1* (*MSP1*) [34] (**Table 1** and **S3**). When we assessed parasite growth in the individual p1-p6 infections (**Fig 4A and 4B**), we found that oocyst size does not vary between parasite isolates in ds*GFP* controls (**Fig 4A and 4B**, ds*GFP* groups). On the other hand, in ds*EcR* females, we detected significant differences in oocyst size between the isolates (**Fig 4A and 4B**, ds*EcR* groups). Despite some inevitable variability due to using different mosquito batches, all isolates increased in size in ds*EcR* females when compared to controls, with the notable exception of p3 in *An. coluzzii* (**Fig 4A**) and p6 in *An. gambiae* (**Fig 4B**). It is important to note that p6 parasites were collected on the same day as p5 and used to infect the same batch of *An. gambiae*

**Table 1. Gametocytemia and COI for *P. falciparum* field isolates.** *P. falciparum* parasites were collected from six different gametocyte carriers (p1-p6 and used for infections in *An. coluzzii* and *An. gambiae* females. Prior to infection, a blood smear was used to count the number of gametocytes present in each sample. Gametocytes were counted per 1,000 leukocytes and then, using an estimated conversion factor of 8,000 leukocytes per µl of blood, converted to gametocytes per µl of blood (Gams/µl). *MSP1*-typing was performed on dried blood spots or recently thawed frozen blood samples and used to determine the (minimum) number of unique parasite genotypes, or complexity of infection (COI), present in each isolate.

| Parasite Isolate | Gams/µl | COI |
|---|---|---|
| p1 | 128 | 3 |
| p2 | 208 | 2 |
| p3 | 120 | 1–3 |
| p4 | 56 | 1–3 |
| p5 | 72 | 3 |
| p6 | 64 | 2 |

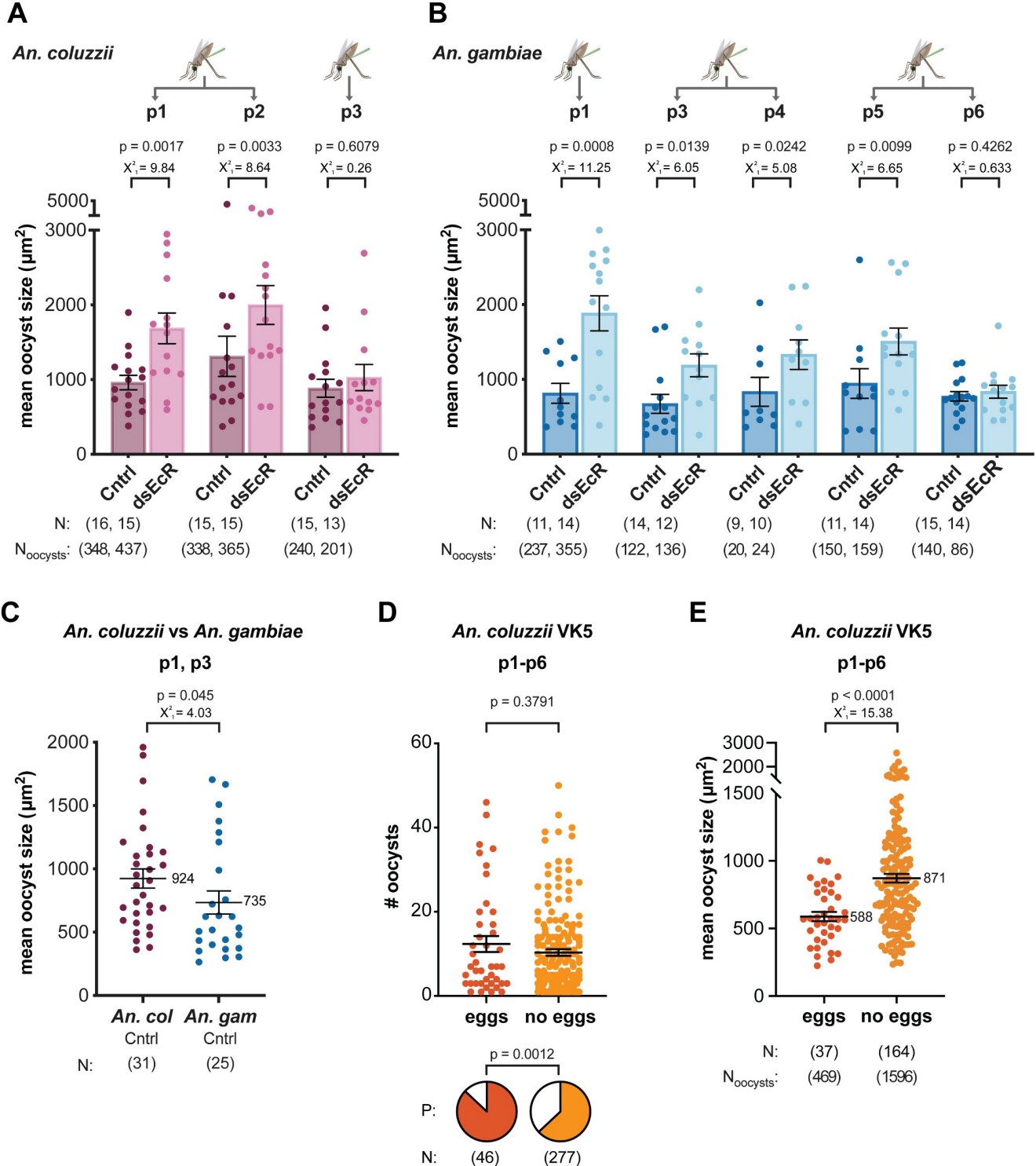

**Fig 4. Individual *P. falciparum* isolates vary in their growth response to *EcR*-silencing. (A)** For *An. coluzzii* females, both p1 and p2 parasites produced larger oocysts in *EcR*-silenced females, but p3 parasites failed to respond to ds*EcR* conditions and did not grow larger than controls (GLMM, LRT). **(B)** For *An. gambiae*, all *P. falciparum* isolates responded to *EcR*-silencing by growing larger except for p6 parasites, which remained the same size as controls (Cntrl), even though they were fed to the same batch of mosquitoes as p5 parasites (GLMM, LRT). **(A-B)** Oocyst size does not vary significantly between parasite isolates in ds*GFP* controls for *An. gambiae* or *An. coluzzii* (*An. coluzzii* ds*GFP*: GLMM, LRT $X^2_2 = 1.52$, p = 0.466; *An. gambiae* ds*GFP*: GLMM, LRT $X^2_4 = 3.47$, p = 0.483), yet it does for ds*EcR*-treated females in both species (*An. coluzzii* ds*EcR*: GLMM, LRT $X^2_1 = 6.3$, p = 0.012; *An. gambiae* ds*EcR*: GLMM, LRT $X^2_1 = 4.67$, p = 0.031). **(C)** Oocyst size varies significantly between *An. coluzzii* and *An. gambiae* control females infected with the same

parasite isolates (p1 and p3) (GLMM, LRT), whereby oocysts are larger in *An. coluzzii* than in *An. gambiae*. **(D)** *An. coluzzii* were collected as larvae from VK5, Burkina Faso and infected with *P. falciparum* isolates p1-p6. Pre-gravid females that failed to develop eggs were less likely to become infected (pie charts, P = oocyst prevalence, Fisher's Exact), although among infected individuals, there was no difference in oocyst intensity (Mann-Whitney). **(E)** Pre-gravid females had significantly larger oocysts at 7 days pIBM compared to gravid females (GLMM, LRT). For all applicable panels, average oocyst size per midgut is shown for simplicity–analyses are based on nested data incorporating all oocyst measurements. N = sample size, or number of mosquitoes. $N_{oocysts}$ = number of individual oocyst measurements. p# = parasite isolate.

mosquitoes, but while p5 showed the characteristic size increase in response to *EcR* silencing, p6 did not (**Fig 4B**). As p5 and p6 differ in both the number and combination of genotypes they contain (**Tables 1 and S3**), these observations suggest that specific parasite genotypes may differ in their interactions with mosquito oogenesis. Interestingly, we observed that p3 parasites did not respond to *EcR* silencing in *An. coluzzii* (**Fig 4A**), yet they increased in size in *An. gambiae* (**Fig 4B**), emphasizing that the same parasite isolate can behave differently in different *Anopheles* species. The only other parasite isolate tested in both species, p1, responded robustly to *EcR* silencing in both vectors. With the caveat that we could perform only two direct comparisons between *An. gambiae* and *An. coluzzii*, these data suggest that the mosquito genotype may also play a role in determining parasite growth rates. Further supporting this idea, in controls infected with p1 or p3 parasites, we see that oocyst growth is significantly different between our two mosquito species, whereby p1 and p3 oocysts grow larger in *An. coluzzii* than in *An. gambiae* (**Fig 4C**).

## Oocyst growth is accelerated in pre-gravid mosquitoes

Given the observed negative association between mosquito oogenesis and parasite growth rates (**Fig 3**), we next set out to investigate oocyst development in pre-gravid mosquitoes (females that fail to develop eggs after taking an initial blood meal), using field-derived *Anopheles* because this phenomenon is highly common in wild populations and considerably less so in laboratory colonies [27,28,35]. To this aim, we collected *An. coluzzii* larvae from natural breeding sites and infected emerged virgin adult females with the same six gametocyte carriers as above (p1-p6). Out of the 323 mosquitoes analyzed in these experiments, the vast majority (86%, N = 277) failed to develop eggs despite taking a blood meal (**S4A Fig**). This effect could be driven by the fact that field mosquitoes are not well-adapted to membrane feeding and may take smaller blood meals, but additionally, it may also suggest that the pre-gravid rate is high in this population as previously observed [27,36].

Altogether, the field mosquitoes averaged 72% infection prevalence with a mean of 11 oocysts per blood feeding event (**S4B Fig**). The pre-gravid females were less likely to become infected than gravid females (pie charts in **Fig 4D**), yet among infected individuals, there was no difference in oocyst intensity (**Fig 4D**). We next determined oocyst size, for practical reasons at 7 days pIBM, one day earlier than in previous experiments. Strikingly, oocysts were significantly larger in pre-gravid females as compared to females that produced an egg batch, with a 48% size increase (**Fig 4E and S4** Table). Therefore, pre-gravid status in *An. coluzzii* has two contrasting effects on parasite infection within the mosquito–it reduces infection prevalence but also strikingly promotes oocyst growth, with probable effects on the EIP.

## Discussion

In this study, we investigated the interplay between mosquito oogenesis and parasite development using multiple *P. falciparum* field isolates and two different *Anopheles* species, including field collected mosquitoes. We found that oocyst size, a good metric for oocyst growth rates [8,26], is negatively linked to oogenesis in *An. coluzzii* and *An. gambiae*, such that disruption

of 20E signaling resulted in accelerated oocyst growth in both species (**Fig 2A–2C**) and across several parasite isolates (**Fig 4A and 4B**). In *An. gambiae*, ds*EcR* treatment also resulted in a greater number of salivary gland sporozoites at the relatively early time point of 12 days piBM (**Figs 2D and S3**). While additional studies assessing sporozoite load across time are still needed to precisely quantify the effects on the EIP, these data confirm that impairing the processes regulating mosquito oogenesis can result in earlier arrival of sporozoites to the salivary glands. Given the lengthy sporogonic cycle of *P. falciparum* (minimum 9 days [6–8,10]) coupled with the relatively limited lifespan of mosquitoes in the field (~2–3 weeks [12–15]), parasites that develop faster will have a greater chance of being transmitted [8]. Although here we could not perform hepatocyte infections due to restrictions in the experimental set-up, our previous laboratory studies showed that faster developing parasites are just as transmissible to human hepatocytes as slower developing parasites [26].

Consistent with accelerated parasite growth in ds*EcR* females, we also found that in both *An. coluzzii* and *An. gambiae* controls, egg development is negatively linked to oocyst size (**Fig 3A and 3B**), confirming that *P. falciparum* growth is influenced by female reproductive processes [26]. In the case of control *An. gambiae*, the impact of oocyst density on this association further suggests that at low-density infections, any alteration in the pathways guiding egg development has a more pronounced effect on oocyst growth compared to high-density infections (**Fig 3C**). Disruption of the 20E signaling pathway perturbs the effect of parasite density on these egg by oocyst size interactions. This time, the most pronounced negative influence of egg number on oocyst growth is observed at high oocyst densities. Proximal mechanisms explaining these results are currently unknown. The relationships among infection, reproduction, nutrient intake, and the overall condition of the host are complex [37,38] and future studies would be required to quantify, for example, the resource levels in control and ds*EcR* mosquitoes with high vs low oocyst densities. Overall, these results demonstrate that there is a strong, conserved interaction between *Anopheles* oogenesis and oocyst growth, but also that 20E signaling regulates this interaction differentially, even in closely related species, indicating that *P. falciparum* interacts uniquely with the post-blood meal environments of these two anophelines. Previously, we demonstrated that the negative association between egg development and oocyst growth is dependent on midgut lipids that the parasite may access via Lipophorin [26]. Given that we observed elevated *Lp* expression in the ds*EcR*-treated females (**Fig 2E**), it is likely that resource allocation and lipids/lipid transporters also play a role in the oocyst growth phenotypes we observed here in the field, although this remains to be tested directly.

Further supporting the link between oogenesis and parasite growth, in infections with field-collected *An. coluzzii*, pre-gravid females that failed to develop eggs had significantly larger oocysts compared to females that produced an egg batch (**Fig 4E**). On the other hand, pre-gravid females were also less likely to become infected initially (pie charts in **Fig 4D**), as also seen previously [20], perhaps because these females take smaller blood meals and therefore are less exposed to infectious parasites. These contrasting effects of the pre-gravid status on two major determinants of vectorial capacity (vector competence and speed of parasite development) make it difficult to conclude how pre-gravid mosquitoes may contribute to malaria transmission dynamics, and models of transmission would be needed to assess this fully. Despite this limitation, given that the pre-gravid condition is quite common and pre-gravid rates over 60% have previously been reported for certain *Anopheles* field populations [27,36], possibly due to low teneral reserves [35,36,39] or environmentally-acquired microbiota affecting nutrient acquisition [40–42], our observations that pre-gravid females favor oocyst growth may suggest that the EIP of parasites in field settings is actually shorter than that typically observed with lab-adapted colony mosquitoes. Furthermore, the effect of pre-gravid status on

oocyst growth rates could be even further exacerbated by these females needing to seek an additional blood meal to complete production of their first egg batch, since it has been shown that a second blood meal significantly accelerates sporozoite formation [8].

In our experiments, oocyst intensity was not affected by *EcR* silencing in either *An. coluzzii* or *An. gambiae* females (**Fig 1E and 1F**). This is in contrast to our previous study where we found a reduction in parasite numbers during the ookinete-oocyst transition. Considering that parasite numbers are roughly 4 times lower here than those achieved in laboratory conditions [26], it is possible that the effects of 20E signaling on parasite survival are dependent on the intensity of infection. Given that oocyst density was also a significant factor affecting egg-oocyst size relationships in *An. gambiae* (**Fig 3C**), it would be worthwhile to further explore how infection load may influence these multiple interactions between parasites and oogenetic processes. Another possible reason for incongruous effects of ds*EcR* treatment on oocyst number may be linked to genetic differences in the parasites and/or mosquitoes used in the study. For instance, in our previous study performed in the laboratory, NF54 parasites and G3 mosquitoes were used, while in the present study it was field-derived mosquitoes infected with naturally circulating parasite isolates. As we found here that mosquito and parasite genetic factors may be linked to differences in oocyst growth responses, it is possible that genetic factors might also influence other interactions between parasites and 20E-regulated processes affecting oocyst number. Interestingly, ds*EcR* treatment did not significantly reduce egg numbers in *An. coluzzii*, while it did in *An. gambiae* (**Fig 1C and 1D**). Due to the low levels of *EcR* expression in females prior to blood feeding, knockdown efficiency was confirmed via increased expression of the lipid transporter *Lp* (one of the genes that is strongly induced by *EcR* silencing [26])(**Fig 2E**), suggesting that silencing was comparable in these species. A possible explanation for these results is that 20E regulation of post-blood feeding processes is slightly different in these two species, as also indicated by the differential effects of ds*EcR* treatment on the egg-oocyst size interaction in *An. coluzzii* compared to *An. gambiae* (**Fig 3A and 3B**). Regardless, disrupting 20E signaling resulted in significantly larger oocysts in both anophelines (**Fig 2A–2C**), demonstrating that 20E regulated pathways have a conserved influence on parasite growth.

While most parasite isolates responded to impaired 20E signaling by accelerating development, the comparison between p5 and p6 in the same batch of *An. gambiae* revealed that p6 parasites did not increase in size (**Fig 4B**). Furthermore, p3 oocysts did not respond to *EcR* silencing in *An. coluzzii* (**Fig 4A**), yet they had a large growth response in *An. gambiae* (**Fig 4B**). Taken together, these results suggest that mosquito-parasite genotype-by-genotype interactions may regulate sporogonic development, as postulated for dengue virus EIP in *Aedes* [43]. This conclusion is further supported by our GLMM analyses, which revealed that mosquito genotype (ds*GFP An. gambiae* vs ds*GFP An. coluzzii*) (**Fig 4C**) and parasite isolate (p1-p6 in ds*EcR* females) (**Fig 4A and 4B**) can significantly influence oocyst growth rates. Considering that high levels of genetic diversity exist in parasite populations in regions with high malaria transmission [44–46] such as Burkina Faso [47], there is likely natural variability in parasite genes regulating development (e.g. genes involved in metabolism, or nutrient transporters). It is plausible that these differences may be caused by the parasite's need to adapt to and be transmitted by different sympatric *Anopheles* species populating the same geographical area [48–50], thus maintaining polymorphisms in genes affecting oocyst growth rates. Identifying the key processes regulating parasite growth and determining the natural polymorphisms in parasite populations in key genes involved in these processes would be a first step to validate this hypothesis, that at present remains highly speculative.

Interestingly, we found that oocyst growth significantly varied across parasite isolates in *EcR*-silenced mosquitoes (*An. gambiae* and *An. coluzzii*) (**Fig 4A and 4B**, ds*EcR*), yet we failed

to detect any isolate-driven effect on oocyst growth in ds*GFP* controls (**Fig 4A and 4B,** Cntrl), including in wild VK5 *An. coluzzii* (**S4C Fig**). This apparent discrepancy may be because the effects of parasite genotype on oocyst growth are only detectable when a great surplus of nutrients is accessible to the parasite throughout development, as when 20E signaling is impaired [26]. Isolate-specific differences in parasite growth may emerge more clearly following additional blood meals, when growth rates are profoundly accelerated [8,22,23]. Additionally, it must be considered that our parasites isolates are largely polyclonal, which may mask differences in parasites growth in normal conditions but not when growth is favored, especially when considering that some genotypes respond differently to the growth stimulus. Providing a second blood meal to these mosquitoes would be most representative of field conditions and may reveal some more subtle differences, as pre-gravid mosquitoes will require a second feed into order to produce eggs.

Altogether, this study expands our knowledge of naturally occurring mosquito-parasite interactions that influence the rate of sporogony in the field. These findings are important for understanding the dynamics of *P. falciparum* transmission in sub-Saharan Africa, and for informing the development of future vector control strategies that seek to interfere with mosquito reproduction. Indeed, our data emphasize that mosquito species may differ in their interactions with parasite isolates. Manipulations to fundamental processes like mosquito reproduction will need to be evaluated for their effects within the specific contexts in which control strategies are implemented and cannot necessarily be extrapolated from laboratory experiments that use single combinations of lab-adapted mosquito and parasite strains.

## Materials and methods

### Ethical approval

Ethical approval was obtained for all procedures involving human subjects from institutional review boards in both the United States and Burkina Faso. Approval through Harvard University was obtained for project #IRB17-0609. This is in agreement with project #18-2018/CEIRES, which was approved by the Institutional Ethics Committee for Research in Health Sciences at the Ministry of Scientific Research, Burkina Faso. All human subjects provided informed consent for participation in the study.

### Mosquito rearing

Experiments were performed with either *An. gambiae* or *An. coluzzii* mosquitoes from independently maintained colonies that were both established in 2008 from gravid females collected in south-western Burkina Faso: Soumousso for *An. gambiae* and Kou Valley for *An. coluzzii* [51]. These colonies are frequently replenished with the F1 offspring of wild-caught, gravid females and species identification is performed using a SINE200 diagnostic PCR [52]. For experiments with field-collected *An. coluzzii*, larvae were collected from breeding sites in VK5, Kou Valley, Burkina Faso and then reared in the laboratory until pupal stages.

Pupae were sex sorted and then allowed to emerge into cages, where adults were maintained on a 10% glucose solution provided *ad libitum* and under standard rearing conditions (27°C, 70–80% humidity, 12h day/night cycle). Female mosquitoes were maintained as virgins throughout adulthood. Prior to blood feeding, mosquitoes were mouth aspirated from cages into large cups sealed with netting and sugar-starved for ~24hr. After blood feeding, engorged mosquitoes were returned to cages and again maintained as before on 10% glucose solution under standard conditions. Before any dissections, mosquitoes were cold-anesthetized and decapitated.

## Gene knockdown using dsRNA

**dsRNA production.** *EcR* (*AGAP029539*) and eGFP fragments were amplified by PCR from the plasmids pCR2.1-EcR and pCR2.1-eGFP (confirmed by DNA sequencing) using universal primers against the pCR2.1 backbone and adding the T7 promoter, as described in [26,53]. Primer sequences were (with T7 promoter in lowercase): pCR2.1-T7F: 5′-taat acgactcactatagggCCGCCAGTGTGCTGGAA-3′; pCR2.1-T7R: 5′-taatacgactca ctatagggGGATATCTGCAGAATTCGCCC-3′. After amplification, PCR products were transcribed into dsRNA using the Megascript T7 transcription kit (Thermo Fisher Scientific) and then purified, as described previously [54]. dsRNA was aliquoted at a concentration of 10 μg/μl and frozen for transport on dry-ice from Boston, USA to Bobo-Dioulasso, Burkina Faso. Prior to experimental use, gel electrophoresis was used to confirm dsRNA quantity and integrity. dsRNA fragments had the following sequences:

eGFP (495 bp):

```
ACGTAAACGGCCACAAGTTCAGCGTGTCCGGCGAGGGCGAGGGCGATGCC
ACCTACGGCAAGCTGACCCTGAAGTTCATCTGCACCACCGGCAAGCTGCC
CGTGCCCTGGCCCACCCTCGTGACCACCCTGACCTACGGCGTGCAGTGCT
TCAGCCGCTACCCCGACCACATGAAGCAGCACGACTTCTTCAAGTCCGCC
ATGCCCGAAGGCTACGTCCAGGAGCGCACCATCTTCTTCAAGGACGACGG
CAACTACAAGACCCGCGCCGAGGTGAAGTTCGAGGGCGACACCCTGGTGA
ACCGCATCGAGCTGAAGGGCATCGACTTCAAGGAGGACGGCAACATCCTG
GGGCACAAGCTGGAGTACAACTACAACAGCCACAACGTCTATATCATGGC
CGACAAGCAGAAGAACGGCATCAAGGTGAACTTCAAGATCCGCCACAACA
TCGAGGACGGCAGCGTGCAGCTCGCCGACCACTACCAGCAGAACA
```

*EcR* (435 bp):

```
CTGCTCCAGTGAGGTGATGATGTTGCGAATGGCCCGCCGGTACGACGCCG
AAACCGACTCCATCCTCTTTGCCAACAACCGATCGTACACGCGCGACTCG
TACAAGATGGCGGGCATGGCGGACACGATCGAGGACCTGCTGCACTTCTG
CCGGCAGATGTACACGCTCACGGTGGACAACGTCGAGTACGCGCTGCTGA
CCGCGATCGTCATCTTCTCCGACCGGCCCGGCCTCGAGAAGGCCGAGCTG
GTGGAAACGATCCAGAGCTACTACATCGACACGCTGCGCGTCTACATCCT
GAACCGGCACGGGGGCGACCCGAAGTGTAGCGTCACGTTCGCGAAGCTGC
TGTCGATCCTGACCGAGCTGCGGACGCTCGGCAACCAGAACTCGGAGATG
TGCTTCTCGCTCAAGCTGAAGAACCGTAAGCTGCC
```

**dsRNA injections.** Adult virgin females <24h post-eclosion were anesthetized on ice and then injected in the thorax with 69 nl of dsRNA (ds*GFP* or ds*EcR*) at a concentration of 10 μg/μl using a Nanoject II (Drummond). Immediately following injection, females were transferred to a large cup for recovery before being moved to a new cage where they were maintained under standard rearing conditions as described above; only females surviving injections (>90%) were used in experiments. 3–4 days post-injection, females were provided with a *P. falciparum*-infected blood meal.

**Gene expression analysis.** *Sample collection*: On the same day that mosquitoes were provided a *P. falciparum*-infected blood meal (3–4 days post-injection), non-blood fed females were collected for gene expression analysis. Females were cold-anesthetized, de-capitated, and then the rest of body samples were stored in 500 μl of RNAlater (Invitrogen). 2 pools of 5 mosquitoes were collected from both ds*GFP* and ds*EcR* groups, with collections done for each batch of mosquitoes that was infected (3 batches of *An. gambiae* and 2 batches of *An. coluzzii* (**Fig 1A and 1B**)). Samples in RNAlater were kept for ~1 day at room temperature or 4˚C, before being frozen at -20˚C. During transport back to Boston, USA from Bobo-Dioulasso,

Burkina Faso, the samples were at room temperature for ~3 days, before again returning to -20˚C for long-term storage.

*RNA extraction & cDNA synthesis*: Using sterile forceps, mosquito samples were removed from the collection tubes containing RNAlater, blotted on a Kimwipe, and then transferred to a fresh set of tubes containing 400 μl of TRI Reagent (Thermo Fisher Scientific). A bead-beater was used to homogenize the samples using two 2-mm beads per tube and beating at 2400 rpm for 2 rounds of 90 seconds each (chilling samples at -20˚C for 5 min between rounds of homogenization). Following homogenization, RNA extraction was performed according to TRI Reagent manufacturer instructions. After extraction, Turbo DNAase (Thermo Fisher Scientific) was used to DNAse treat the samples. NanoDrop Spectrophotometer 2000c (Thermo Fisher Scientific) was used both before and after DNAase treatment to assess the quality and quantity of RNA. About 3 μg of RNA was used for cDNA synthesis, performed as described in [55].

*qRT-PCR*: *EcR* expression is very low in non-blood fed females and unreliable for direct assessment of gene knockdown efficiency; therefore, transcript levels of the mosquito lipid transporter *Lipophorin* (*Lp*) were used as a reporter for *EcR* knockdown, because *Lp* is known to be strongly induced by *EcR* silencing independent of blood feeding [26] (**Fig 2E**). To this end, quantitative real-time PCR (qRT-PCR) was performed on a QuantStudio 6 Pro thermocycler (Thermo Fisher Scientific) in 15 μl reactions containing 1X PowerUp SYBR Green Master Mix (Thermo Fisher Scientific), primer dilutions, and 5 μl of sample cDNA diluted 1:10. Primers (Integrated DNA Technologies) targeting *Lipophorin* (*Lp*) (*AGAP001826*) (Fwd: CAGCCAGGATGGTGAGCTTAA; Rev: CACCAGCACCTTGGCGTT) or the ribosomal gene *RpL19* (*AGAP004422*) (Fwd: CCAACTCGCGACAAAACATTC; Rev: ACCGGCTTCTTGATG ATCAGA), used as a housekeeping control, were used in our analysis. Relative gene expression was determined using the delta-delta-Ct method, normalizing *Lp* expression against *RpL19*, and ds*EcR* samples against ds*GFP*.

## Infections with *P. falciparum* field isolates

**Identification of gametocyte carriers.** *P. falciparum* gametocyte carriers were identified in the villages surrounding Bobo-Dioulasso, as described in [56,57]. Briefly, children 5–13 years old were surveyed for the presence of *P. falciparum* parasites by fingerprick blood smears, which were analyzed by Giemsa staining and light microscopy. Children of this age range were screened because it has been demonstrated that individuals between 5–15 years old exhibit significantly higher gametocyte prevalence and density compared to individuals older than 15 years [58,59]. Subjects were asked if they would participate in the study if they were found to have a gametocytemia between 5–40 gametocytes per 1000 leukocytes (or 40–320 gametocytes/μl blood, based on an estimated conversion factor of 8000 leukocytes/μl blood). All persons found to be infected with malaria from this screen were offered treatment regardless of enrollment in the study.

**Membrane feeding.** If parental consent was granted, gametocyte carriers were brought to the lab where < 6 ml of venous blood was drawn by a trained technician. Blood was spun down at 1800 rpm for 5 min at 37˚C to separate the red blood cells from the serum, and then the sample serum was removed and replaced with purchased serum from persons naïve to malaria [56,60]. After resuspension, the blood was introduced into 20–30 glass membrane feeders heated to 37˚C with a Haake water pump. *An. gambiae* or *An. coluzzii* (field-derived or VK5 collected) were contained in large cups and allowed to feed through a parafilm membrane sealing the bottom of the glass feeders for 60–120 mins [57,60,61]. Remaining sample blood was maintained at 37˚C and supplemented into the feeders as needed. Once feeding time was

complete, any unfed or partially-fed females were mouth aspirated and removed from the experiment. Six different parasite isolates, collected from six different gametocyte carriers (denoted as p1-p6), were used for our experiments. Three isolates were used to infect *An. coluzzii* colony mosquitoes (p1-p3), and five were used to infect *An. gambiae* colony mosquitoes (p1, p3-p6) (**Fig 1A and 1B**). All six isolates were used in infections with VK5 *An. coluzzii* (p1-p6). Although we had hoped to infect all colony mosquitoes with all six parasite isolates, this was not possible due to the inherent limitations of working with field-derived mosquitoes and human-collected parasite isolates. Importantly, gametocyte-containing blood does not remain infectious and cannot be stored for use after the morning of collection. Therefore, we performed our infection experiments using the mosquitoes that were available at the time of the blood collection.

## Assessing fecundity and pre-gravid status

Females were intentionally maintained as virgin throughout experimentation to eliminate the variable of mating status on our study of mosquito-parasite interactions. Furthermore, because females were virgin, oviposition did not occur following blood feeding and fecundity/pre-gravid status could be determined at the same time that *P. falciparum* midgut infection was assessed 7–8 days pIBM. At this time, ovaries were dissected and developed eggs were counted (dsRNA-injected *An. gambiae*, *An. coluzzii*) or females were scored as gravid/pre-gravid (VK5 *An.* coluzzii). Assessing egg development in this way concurrently with midgut dissections allowed for paired egg-oocyst data collection.

## Assessing *P. falciparum* infection in the midgut

7 days (VK5 *An.* coluzzii) or 8 days (dsRNA-injected *An. gambiae*, *An. coluzzii*) post-infected blood meal, female midguts were dissected and stained in 1% mercurochrome. Light microscopy was then used to count the number of *P. falciparum* oocysts present in the midgut, determining the prevalence and intensity of infection. Images of the infected midguts were also obtained, and from these images oocyst size was determined by manually measuring the cross-sectional area of all intact oocysts using FIJI software [62]. Any blurry midgut images, for which oocyst measurements could not be reliably determined, were excluded from our analysis.

## Assessing sporozoites in the salivary glands

Individual female salivary glands were dissected into 1X phosphate-buffered saline (PBS) at 12 days pIBM. The salivary glands were hand homogenized using a plastic pestle and spun at 8000 $g$ for 10 min. The supernatant was discarded, and the remaining pellet (containing sporozoites) was resuspended in PBS to a known volume. Phase-contrast microscopy was used to count sporozoites present in 0.1 µl using disposable hemocytometers. These counts were then used to calculate the total number of sporozoites present in the salivary glands of each mosquito. The lower limit of detection for an individual female was ~120 sporozoites.

## Genotype analysis of *P. falciparum* field isolates

**Collection of blood spots and frozen blood samples.** Immediately following collection from gametocyte carriers, a small volume of blood was spotted onto Whatman paper to generate dried blood spots. Additionally, a small volume of the blood (~1 ml) was added to Glycerolyte 57 (Baxter Fenwal 4A77831) at a 1:1 ratio and aliquoted into Nunc cryotubes. The samples

were kept at -80˚C until they were shipped on dry-ice back to Boston, USA and transferred to liquid nitrogen for long-term storage.

**Parasite DNA extraction & confirmation of *P. falciparum* infection.**   DNA was extracted from dried blood spots or recently thawed frozen blood samples (in the case of p5 and p6) using a DNeasy Blood and Tissue kit (Qiagen), involving lysis at 56˚C, column purification, and elution. Extracted DNA was then used for a nested PCR of 18S rDNA according to Snounou *et al* [63] to confirm that only *P. falciparum* parasites, and no other human *Plasmodium* species, were present in the samples. Primers for the first PCR reaction were:

rPLU-F: 5′-TTAAAATTGTTGCAGTTAAAACG-3′;
rPLU-R: 5′-CCTGTTGTTGCCTTAAACTTC-3′.

The subsequent, nested PCR reactions used primers specific for *P. falciparum*, *P. malariae*, or *P. ovale* with the following sequences, respectively:

rFAL-F: 5′-TTAAACTGGTTTGGGAAAACCAAATATATT-3′;
rFAL-R: 5′-ACACAATGAACTCAATCATGACTACCCGTC-3′.
rMAL-F: 5′-ATAACATAGTTGTACGTTAAGAATAACCGC-3′;
rMAL-R: 5′-AAAATTCCCATGCATAAAAAATTATACAAA-3′.
rOVA-F: 5′-ATCTCTTTTGCTATTTTTTAGTATTGGAGA-3′;
rOVA-R: 5′-GGAAAAGGACACATTAATTGTATCCTAGTG-3′.

Gel electrophoresis was used to evaluable all PCR products and confirm that only *P. falciparum* parasites were present in our samples (p1-p6).

***MSP1*-typing and COI determination.**   Extracted sample DNA was used to determine the *MSP1* genotype(s) present in each sample by nested PCR as described by [34]. This nested PCR approach was used to identify the allelic type of 4 variable regions on the *MSP1* gene termed block 2, 4A, 4P, and 6. Each block can either have a MAD20 or K1 allelic family type, with the exception of block 2, which can alternatively have an RO33 type. The first PCR reaction was used to type blocks 2 and 6. To do this, the following primers (3 forward; 2 reverse) were used in six unique combinations for each parasite isolate:

M2F: 5′-GGTTCAGGTAATTCAAGACGTAC-3′
K2F: 5′-TCTTAAATGAAGAAGAAATTACTACAAA-3′
R2F: 5′-TAAAGGATGGAGCAAATACTCAAGT-3′
M6R: 5′-ATTTGAACAGATTTCGTAGGATCTTG-3′
K6R: 5′-GATATTCATTGGGTTCAGTTGATTC-3′

PCR products were then run on a gel in order to identify blocks 2 and 6 as: M—M, M—K, K—M, K—K, R—M, or R—K. PCR products from the first reaction were then used in subsequent nested PCRs where blocks 4A and 4P were typed. This second round of PCRs was done using the following primers (2 forward; 2 reverse) in four unique combinations:

M4AF: 5′-TTGAAGATATAGATAAAATTAAAACAGATG-3′
K4AF: 5′-AATGAAATTAAAAATCCCCCACCGG-3′
M4PR: 5′-TCGACTTCTTTTTTCTTATTCTCAG-3′
K4PR: 5′-TCCTCGATTTTTTTGTTCTTATCAAG-3′

The resulting PCR products were run on a gel, identifying blocks 4A and 4P as: -MM-, -MK-, -KM-, or -KK-. Combining the typing results from all four blocks (*i.e.* MMMM, KMMK, RMKK, etc), allows for the identification of up to 24 distinct *MSP1* genotypes. These nested PCR reactions were performed on blood sample DNA (p1-p6) to determine which *MSP1* genotypes were present in each isolate (**S3 Table**). The number of unique *MSP1* genotypes identified in each sample was then used to determine the sample's (minimum) complexity of infection (COI) (**Table 1**). NF54 cultured parasites were used a control for these experiments, as they are clonal and known to contain only a single parasite genotype.

### Statistical analyses

GraphPad Prism 8, JMP Pro 14, and R were used to analyze the data with a significance cut-off of p = 0.05. Oocyst size data was analyzed using Generalized Linear Mixed Effects Models (GLMM) and Likelihood Ratio Tests (LRT), allowing for nesting of individual oocyst measurements by mosquito, and assessment of both fixed and random effects. Other continuous data (egg, oocyst, and sporozoite number) was first analyzed for normality (D'Agostino/Pearson and KS normality test) and transformed to generate normally distributed data when possible. If normally distributed, data was then analyzed by either unpaired t-test or one-way ANOVA (Welch's correction when appropriate). Non-normally distributed data was analyzed by either Mann-Whitney or Kruskal-Wallis. Prevalence data (for oocysts and sporozoites) was analyzed using Fisher's Exact test.

Specific statistical tests and p values are indicated in the text, figures, and figure legends. All graphs display mean values with standard error of the mean (SEM).

## Supporting information

**S1 Fig. Egg development for individual infections. (A-B)** The effects of *EcR*-silencing on egg development across individual infections for **(A)** *An. coluzzii* (unpaired t-test and Mann-Whitney) and **(B)** *An. gambiae* (unpaired t-test and Mann-Whitney) compared to controls (Cntrl). N = sample size. p# = parasite isolate.
(TIF)

**S2 Fig. Oocyst prevalence and intensity for individual infections. (A-B)** The effects of *EcR*-silencing on the prevalence (Fisher's Exact) and intensity (unpaired t-test and Mann-Whitney) of oocysts across individual infections for **(A)** *An. coluzzii* and **(B)** *An. gambiae*, compared to ds*GFP*-injected (Cntrl) females. P = oocyst prevalence. N = sample size. p# = parasite isolate.
(TIF)

**S3 Fig. Sporozoite prevalence and intensity for individual infections. (A)** The salivary glands of ds*EcR* and ds*GFP* (Cntrl) *An. gambiae* females were assessed at 12 days post-infected blood meal from infections with parasite isolates p1, p3, p5, and p6. The sporozoite prevalence (Fisher's Exact) and intensity (Mann-Whitney) for each infection are shown. P = sporozoite prevalence. N = sample size. p# = parasite isolate.
(TIF)

**S4 Fig. Additional infection data for colony mosquitoes and VK5-*An. coluzzii*. (A)** The majority of VK5-*An. coluzzii* females that were provided a parasite-infected (p1-p6) blood meal, failed to develop any eggs. **(B)** Oocyst prevalence (P) and intensity for individual infections with VK5-*An. coluzzii*. **(C)** Mean oocyst size per VK5-*An. coluzzii* female for each infection with a different parasite isolate (P#). Mean oocyst sizes are shown for simplicity, but all analyses were done with all individual oocyst measurements nested by mosquito. N = sample size. P# = parasite isolate.
(TIF)

**S1 Table. Full-factor GLMM Output for *An. coluzzii*.** Effect of treatment (ds*GFP* vs ds*EcR*), oocyst number, egg number, and their interactions on oocyst size. In this model, treatment, oocyst number and egg number were considered as fixed effects whereas parasite isolate and mosquito ID were set as random effects. Significant effects are in bold.
(XLSX)

**S2 Table. Full-factor GLMM Output for *An. gambiae*.** Effect of treatment (ds*GFP* vs ds*EcR*), oocyst number, egg number, and their interactions on oocyst size. In this model, treatment,

oocyst number and egg number were considered as fixed effects whereas parasite isolate and mosquito ID were set as random effects. Significant effects are in bold.
(XLSX)

**S3 Table. Additional information on *P. falciparum* isolates.** *MSP1*-genotyping results for *P. falciparum* isolates (p1-p6) used in infections with *An. gambiae* (*An. gam*), *An. coluzzii* (*An. col*), and/or VK5-collecte*d An. coluzzii* (*VK5-An. col*) (See also **Methods**).
(XLSX)

**S4 Table. Full-factor GLMM Output for VK5 *An. coluzzii*.** Effect of egg development (eggs vs no eggs), oocyst number, and their interactions on oocyst size. In this model, egg development and oocyst number were considered as fixed effects whereas parasite isolate, mosquito generation, and mosquito ID were set as random effects. Significant effects are in bold.
(XLSX)

## Acknowledgments

We wish to thank all the students and technicians at the IRSS/IRD who provided valuable assistance for the experiments of this study. We especially wish to thank Hélène Somé, Baudoin Dabiré and Fulgence Da for mosquito rearing, Charles Nignan for coordinating VK5 larval collections, and Souleymane Tamboula for assistance removing unfed females following infectious feeds. We also wish to thank Perrine Marcenac for helpful discussions of the data and Manuela Bernardi for graphical artwork. Finally, we'd like to thank the local villagers and study participants.

## Author Contributions

**Conceptualization:** Kristine Werling, Maurice A. Itoe, W. Robert Shaw, Thierry Lefèvre, Flaminia Catteruccia.

**Data curation:** Kristine Werling, Maurice A. Itoe, W. Robert Shaw, Kelsey L. Adams, Duo Peng, Thierry Lefèvre.

**Formal analysis:** Kristine Werling, Thierry Lefèvre, Flaminia Catteruccia.

**Funding acquisition:** Flaminia Catteruccia.

**Investigation:** Kristine Werling, Maurice A. Itoe, W. Robert Shaw, Raymond Dombagniro Hien, Bali Jean Bazié, Fofana Aminata, Kelsey L. Adams, Bienvenu Seydou Ouattara, Mathias Sanou, Dari F. Da, Rakiswendé Serge Yerbanga, Thierry Lefèvre, Flaminia Catteruccia.

**Methodology:** Kristine Werling, Dari F. Da, Rakiswendé Serge Yerbanga, Thierry Lefèvre, Flaminia Catteruccia.

**Project administration:** Kristine Werling, Maurice A. Itoe, W. Robert Shaw, Dari F. Da, Rakiswendé Serge Yerbanga, Thierry Lefèvre, Flaminia Catteruccia.

**Resources:** Roch K. Dabiré, Dari F. Da, Rakiswendé Serge Yerbanga, Abdoulaye Diabaté, Thierry Lefèvre, Flaminia Catteruccia.

**Software:** Kristine Werling, Duo Peng, Thierry Lefèvre.

**Supervision:** Kristine Werling, Maurice A. Itoe, W. Robert Shaw, Roch K. Dabiré, Abdoulaye Diabaté, Thierry Lefèvre, Flaminia Catteruccia.

**Validation:** Kristine Werling, Thierry Lefèvre, Flaminia Catteruccia.

**Visualization:** Kristine Werling, Thierry Lefèvre, Flaminia Catteruccia.

**Writing – original draft:** Kristine Werling, Maurice A. Itoe, W. Robert Shaw, Kelsey L. Adams, Thierry Lefèvre, Flaminia Catteruccia.

**Writing – review & editing:** Kristine Werling, W. Robert Shaw, Thierry Lefèvre, Flaminia Catteruccia.

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
