## [Decision Letter · Decision Letter 0]

13 Oct 2023

Dear Dr Catteruccia,

Thank you very much for submitting your manuscript "Impairing reproductive processes in *Anopheles* vectors accelerates development of circulating isolates of *Plasmodium falciparum* malaria" for consideration at PLOS Neglected Tropical Diseases. As with all papers reviewed by the journal, your manuscript was reviewed by members of the editorial board and by several independent reviewers. The reviewers appreciated the attention to an important topic. Based on the reviews, we are likely to accept this manuscript for publication, providing that you modify the manuscript according to the review recommendations. 

Sincerely,

Mariangela Bonizzoni

Academic Editor

Abhay Satoskar

Section Editor

Reviewer's Responses to Questions

**Key Review Criteria Required for Acceptance?**

**Methods**

-Are the objectives of the study clearly articulated with a clear testable hypothesis stated?

-Is the study design appropriate to address the stated objectives?

-Is the population clearly described and appropriate for the hypothesis being tested?

-Is the sample size sufficient to ensure adequate power to address the hypothesis being tested?

-Were correct statistical analysis used to support conclusions?

-Are there concerns about ethical or regulatory requirements being met?

Reviewer #1: No concerns.

Reviewer #2: Study design is appropriated and the data supports the hypothesis presented. I don't see any statistical issues in the manuscript.

Reviewer #3: Objectives clearly articulated. I think in some cases the power is lacking to address some of the hypotheses but this is a restriction that comes with this type of study, rather than poor design. I have commented on the text for such occasions.

**Results**

-Does the analysis presented match the analysis plan?

-Are the results clearly and completely presented?

-Are the figures (Tables, Images) of sufficient quality for clarity?

Reviewer #1: Generally well-presented, only minor comment regarding the figures to improve clarity

Reviewer #2: The hypothesis is well supported by the dat, the results are clear. Figures and diagrams are very well put together.

Reviewer #3: The figures are of good quality and are explanatory. The results are hard to follow but again, this is because they are complex and sometimes contradictory, depending on either genotype, batch or condition. I think a schematic figure summarising findings might be helpful.

**Conclusions**

-Are the conclusions supported by the data presented?

-Are the limitations of analysis clearly described?

-Do the authors discuss how these data can be helpful to advance our understanding of the topic under study?

-Is public health relevance addressed?

Reviewer #1: The authors tackle an important question as to how lab observations equate to more natural infection settings. The manuscript can be improved by better articulating how potential differences may result between this study and those previously published from the group.

Reviewer #2: In the present manuscript the authors investigate if that relationship between reproduction and malaria transmission is relevant to endemic regions that present different parasite genotypes and mosquito species. The work is very relevant and well put together.

The differences found between colluzzii and gambiae are very interesting and point that the role of 20E might not be only related to oogenesis in gambiae.

Very interesting the data with the pre-gravid females collected from the field. I wondered if the presumably different microbiota from the field collected and the lab colonies could play a role in the percentage of pre-gravid females. Please address this in the discussion.

Reviewer #3: Yes i think the conclusions are supported by the data presented and the public health relevance is addressed. Limitations of the analysis and caveats on the interpretation are usually well considered, although there is the impression that the language leans to emphasise one interpretation over another when the data might be less conclusive - for example 'there is a trend' when not significant but in line with the preferred hypothesis and 'no effect' when not significant and not in line with preferred hypothesis (see also 'partially contrast').

**Editorial and Data Presentation Modifications?**

Reviewer #1: (No Response)

Reviewer #2: Line 41 – “when we disrupt signally” - replace by signaling.

Line 62 – phrase reads strange to me – replace by “female feeds on blood again”.

Reviewer #3: I find the short title provided by the authors "Plasmodium growth is linked to mosquito oogenesis in field settings" to be a fairer, and more accurate reflection of the experimental findings presented in this paper compared to the current working title. I would ask the authors to consider using the shorter form, or a modified version thereof.

**Summary and General Comments**

Reviewer #1: (No Response)

Reviewer #2: Previously this group had reported that mosquito reproduction is negatively correlated to P.falciparum oocyst growth rates. That implies that mosquitoes that do not develop eggs can transmit parasites sooner, representing an advantage to the parasite considering the lifespan of the mosquito. I have some small comments and I believe minor editing should be done before publication, but the manuscript is well written, the hypothesis is supported by the data and the work represents an important advancement in our understanding of the trade of between reproduction and parasite growth in the field.

Reviewer #3: This is an important piece of work, looking at the interaction between oogenesis and parasite development. While it has long been known that there are trade offs in investment in immunity, which can affect parasite development and numbers, and reproduction, these authors previously showed in lab mosquitoes that the correlation was not necessarily as expected; they interfered with oogenesis by disrupting ecdysone signalling and this resulted in faster oocyst development.

In this study they expand this study to field caught (or progeny of field caught mosquitoes) to see how true this holds in nature. Not all the previous findings can be recapitulated, which is fine and some what to be expected - if things always held true there would be no point testing in the field first. These findings will be of interest to a wide range of researchers in the field of vector control and insect physiology. As mentioned above, I do think there are some sections where findings are a little overinterpreted and the language could be modified. Some of the generality of the findings is limited in power by the availability of all parasite and vector genotype combinations but this is still a significant amount of work and commendable. 

Rather than list all my points here I have included an annotated copy of the manuscript and below i list just a few of the more significant technical comments/queries:

There is a general incongruity in the level of statistical description and/or language that is used in the different sections

Line 199 you still cannot exclude batch effects (on the parasite side) here though, can you?

presumably you don't have separate batches of the parasite? And p6 was not compared across mosquito batches, or across mosquito species, was it?

Line 247 re: Fig4E oocyst sizes in pre-gravid versus gravid: this seems confounded by the fact that An col and An gamb are mixed here. An gam apparently had zero 'pre-gravid' females, so there can be no signal here. If you remove An gambiae and look only at An coluzzi, is there a significant difference?

Line 274 re: interactions at low density vs high density: this seems circular to me. Accepting, for a minute, the hypothesis of limiting available resources - the relative decrease per oocyst is the same in both condition. In those where there is higher density of oocysts and therefore even less available resources to start with, wouldn't one expect the effect to be further exacerbated at this density. The point made here seems stretched and thin.

Line 278 'perturbs the effect of parasite density on egg-oocyst interactions' there is a lot in this statement - at first reading one could take this to mean there is no longer an effect but in fact, the effect of density is still there, but the order of strength of correlation at the different densities is now reversed, is it not? i.e. in the control condition the negative correlation is highest (slope is steepest) at low densities, whereas in the dsEcR condition the negative correlation is highest at high densities.

Line 324 (and elsewhere) Re: lack of ability to validate EcR knockdown after dsEcR injection. It would would be nice to get more clarity on this (here or in Methods). Could you not look at EcR transcript levels after bloodfeeding? Did you look at EcR transcript levels and were not able to detect transcripts? Or able to detect transcripts but no knockdown?

Line 344 re: need to adapt to sympatric species. This is a lot of speculation and, interesting as it is, has little data to support. You might at least suggest for the reader the next steps to prove it

Line 351 this is a great point, but doesn't it argue against the hypothesis just made (344-347)?

Line 459 out of interest, what was the justification for only taking children?

PLOS authors have the option to publish the peer review history of their article (what does this mean?). If published, this will include your full peer review and any attached files.

Reviewer #1: No

Reviewer #2: No

Reviewer #3: No

Figure Files:

Data Requirements:

Reproducibility:

References

---

## [Editor Report · Decision Letter 1]

28 Dec 2023

Dear Dr Catteruccia,

We are pleased to inform you that your manuscript 'Development of circulating isolates of *Plasmodium falciparum* is accelerated in *Anopheles* vectors with reduced reproductive output' has been provisionally accepted for publication in PLOS Neglected Tropical Diseases.

Best regards,

Mariangela Bonizzoni

Academic Editor

Abhay Satoskar

Section Editor

---

## [Editor Report · Acceptance letter]

5 Jan 2024

Dear Dr Catteruccia,

We are delighted to inform you that your manuscript, "Development of circulating isolates of *Plasmodium falciparum* is accelerated in *Anopheles* vectors with reduced reproductive output," has been formally accepted for publication in PLOS Neglected Tropical Diseases.

Best regards,

Shaden Kamhawi

co-Editor-in-Chief

Paul Brindley

co-Editor-in-Chief
